# A sustainable lithium bromide-water absorption cooling system using automobile engine waste heat: Theoretical study

**Mohammed Qasim Shaheen**[ID]*, **Salman Hashim Hammadi**

Department of Mechanical Engineering, College of Engineering, University of Basrah, Basrah, Iraq

* engpg.mohammed.qasim@uobasrah.edu.iq

## Abstract

This manuscript investigates the utilization of waste heat from automobiles, such as exhaust gas and engine coolant water, to run a lithium bromide water Absorption Cooling System (ACS). This study proposed adding a secondary heat exchanger located between the primary heat exchanger and generator. It takes heat from engine coolant water to reduce thermal load on the generator and to enhance the Coefficient of Performance (COP) of the system. The effect of concentration solution, primary heat exchanger effectiveness, and the temperature of main component (generator, condenser, evaporator and absorber) are studied. The results show a COP of 0.79 with a cooling capacity of 5 kW at generator, condenser, evaporator, and absorber temperatures of (90, 40, 10, and 35), respectively. The COP increases as the evaporator temperature increases, and it decreases as the condenser and absorber temperature increases. Also, any increase in the heat exchanger effectiveness will be led to increase the COP. The results show that the addition of a secondary heat exchanger led to reduce the load on the generator by 4% to 7%, and that depends on the operating conditions and the system. In addition, the results examine a significant reduction in $CO_2$ emissions by 1.58 kg/hr. These findings point out to a substantial possibility for lowering thermal emissions and increasing energy efficiency, and that provides a long-term way to use waste energy in automobiles.

## 1. Introduction

The vapor compression cycle is the common employed cooling method in vehicles today. However, the refrigerants used in vapor compression refrigeration systems are primarily hydrocarbons, such as HCFCs and HFCs, which are not environmentally friendly [1,2]. These types of refrigerants are polluting the environment as well as the atmosphere, and that cause a global warming and depletion of the ozone layer [3–5]. Furthermore, the system places additional demands on the engine to ensure that the engine shaft power is sufficient to run the condenser in the compression system.

**Data availability statement:** All relevant data are within the paper.

**Funding:** The author(s) received no specific funding for this work.

**Competing interests:** The authors have declared that no competing interests exist.

As a result, excessive effort and energy are produced, which builds up and causes unfavorable changes in the environment [6]. It is widely recognized that an IC engine has an efficiency of approximately 35% to 40% [7]. This implies that only one-third of the energy generated by the combustion of the fuel is converted into useful work, specifically mechanical output, and approximately 60–65% of the energy is lost to the environment in the form of heat [8–10]. Around 28%-30% of the energy is lost through coolant and lubrication losses, 30%-32% is lost through exhaust gases from the exhaust pipelines, and the remaining energy is lost through radiation and convection [11]. Consequently, the development of absorption devices is crucial for both air conditioning and preservation applications [12,13].

The objective of this study is to analyze and evaluate an ACS in which automotive engine waste heat acts as the energy source. This have been done by incorporating an additional heat exchanger between the existing heat exchanger and the generator, which uses engine coolant heat. The efficiency and the performance of ACS is investigated at different operating conditions and solution concentration in terms of temperature variation in the main components. Also, the system is assessed in terms of its environmental benefits, such as emissions reduction and potential sustainability benefits which is arising from reduced fuel consumption.

## 2. Literature review

Vicatos et al. [14] are considered a Nissan 1400 compact truck engine-exhaust-driven air conditioning system. Since their COP values were low, 0.08, they are concluded that their design needs to have a considerable improvement. They adapt Vapour Absorption Refrigeration System (VARS) units in their work, in which ammonia–water ($NH_3 - H_2O$) are taken as the working pairs of fluids. The cooling load for their design was acceptable at 0 C, which is estimated to be 2 kW. Manzela et al. [15] was studied the $NH_3 - H_2O$ in VARS which is powered by the exhaust of an internal combustion engine. The system was evaluated at a different throttle valve opening, where the system achieves a steady-state temperature, which is ranging from 4°C to 13°C after three hours. Mohapatra et al. [9] were suggested a 2.8-liter V6 diesel engine as a potential power source for a 10.55 kW absorption chiller, which can be powered by engine exhaust emissions. Rêgo et al. [16] were investigated, experimentally, $NH_3 - H_2O$ of VARS by using the exhaust gases of the internal combustion engine. The exhaust stream had enough heat to power the VARS between 1500 and 4300 rpm. Ther is a favorable decrease in the evaporation temperature at the engine speed between 1500 and 2500 rpm Hilali and Soylemez. [17] were built an engine of exhaust-powered $LiBr - H_2O$ of VARS, that can produce 2.5 kW. The COP was 0.78 at an evaporator temperature of 11°C. The VARS undergoes better COP when the heat recovered since the exhaust is less than 9 kW. Aly et al. [18] were used the exhaust from a diesel engine to power a diffusion absorption refrigeration (DAR) system. Their experimental results show that the DAR system could control exhaust gas passage over a wide range of engine load. Depending on engine load, the chilled cabin reached to 10°C –14.5°C after 3.5 hours of system activation, where a controlled exhaust flow rate provides (30 Nm) torque which had the best COP of 0.10.

Adjibade et al. [19] had an experimental study on DAR cycle, in which they used a heat exchanger to extract the heat from the exhaust gases of an internal combustion engine, or using an electric grid as a source of heat. The operating temperatures of the exhaust and the electrical grid were 140°C to 152°C. They concluded that exhaust gases can power the DAR cycle. Yuan et al. [20] proposed a VARS to recover heat from marine engine exhaust gasses. They used a ternary ($NH_3$–$H_2O$–$LiBr$) mixture for the system. Compared to the binary system, under refrigeration conditions below -15.0°C, the ternary system operates. Kaewpradub et al. [21] investigated a Single-effect $LiBr - H_2O$-based VARS on a 4-stroke gasoline engine exhaust. Engine operating between 1200 rpm to 1600 rpm which is ideal for running the VARS. It was found that the speeds greater than 1600 rpm promote $LiBr - H_2O$ solution crystallization as a result of the increase in the generator temperature. At 1400 rpm, the COP reached 0.275 and a cooling load of 700 W. Sharma et al. [22] utilized a hot water-driven single-stage $LiBr - H_2O$ solution of ACS to investigate and assess the impact of various factors on the COP. As the evaporator pressure increases, the condenser pressure increases, and that lead to decrease the COP. However, the COP increases when the concentration of the $LiBr$ solution in the absorber decreases. Táboas et al. [23] employed a range of absorbents and $NH_3$, as a refrigerant, to evaluate the VARS, which is powered by jacket water in fishing vessel engines. Simulations show that $NH_3/(LiNO_3 + H_2O)$ and $NH_3/LiNO_3$ have greater COP values than $NH_3/H_2O$. At 25 °C condensing temperature, the minimum evaporation temperatures of the working fluid cycles of $NH_3/LiNO_3$, $NH_3/(LiNO_3 + H_2O)$ and NH3/H2O were the lowest at -18.8 °C, -17.5 °C, and -13.7 °C, respectively. Palomba et al. [24] proposed onboard absorption device to test a 195 kW marine engine. The results show an annual reduction in the fuel consumption about 1600 kg, and a reduction in the $CO_2$ emissions by three tones. Kanase [25] suggested an ACS rather than a compression cooling system, which is more stressful on engines and worse for the environment. Cooling water and wasted engine exhaust gases were utilized. Because ACS uses waste heat from the engine, it was a successful substitute for air conditioning in cars. Cao et al. [26] proposed a cascaded absorption-compression cycle as a solution for the refrigeration and cooling requirements of a cargo ship. The simulation testing shows a system that is powered by the heat of the exhaust lead to a significant decrease, in both, the $CO_2$ emissions by 11% and petroleum consumption by 38% than the existing system. Ammar and Seddiek [27] examined the VARS using jacket cooling solutions and shipboard diesel engine exhaust gases from a thermo-economic and environmental perspective. The VARS can reduce the annual fuel usage by 156 tones and emissions by 6.3% as compared to the main engine during a cruise. Sreenesh Valiyandi [28] designed an eco-friendly DAR by investigating six dis-similar types of refrigerant blends. The study established that the optimum system performance was attained when using the isobutane + dimethylformamide + helium mixed refrigerant. Leyla et al. [29] integrates a solid oxide fuel cell with refrigeration and electrolysis to produce power, cooling, heating, and hydrogen from natural gas. Decreasing efficiency and increasing $CO_2$ emissions with current density is increased, while genetic algorithm optimization yields ability to maximize exergy efficiency of 0.6443. Mehmet Erhan [30] Integrated an ACS with an autoclave to utilize waste steam heat for cooling. Using an ammonia/water cycle, a cooling load of 130.9 W was achieved with 531.7 W of waste heat, resulting in a COP of 0.246. This system enhances autoclave energy efficiency and preserves medical supplies. Kadam et al. [31] compared the performance of several variations of refrigeration cycles based on vapor compression and absorption with conventional and innovative working fluids such as $NH_3 - H_2O$ and acetaldehyde-N,N-dimethylformamide. The results indicate that the cascade system with acetaldehyde N, N-dimethylformamide generates cooling efficiency of up to 67.8% lower than conventional systems. The results show that there is a decrease in the cost and the global warming potential of up to 83.3% and 67.8%, respectively. Additionally, the electrical energy consumption in this system is reduced to 30.5%. Thampi [32] studied the DAR to reduce emissions and enhance the performance. The possible alterations to the system are, hybrid working fluids, nano particle additives, and ionic liquids. Among refrigeration systems, it also mentions that 17 percent of electrical energy is consumed Thus, the research is being directed into improving the performance, but not reducing the costs and energy consumption. Huang et al. [33] suggested using waste heat in liquid cooling data centers with compression-assisted absorption refrigeration heating technology. The system can cool 4.29 kW and heat 1.37 kW. The result showed that raising the compression ratio

from 1.067 to 2.0 lowers the minimum generation temperature from 52.0°C to 27.2°C. Flow split ratios and circulation ratios are quantified to help optimize refrigeration and heating loads.

 The review of the previous studies in this field has shown that many researchers have investigated the feasibility of operating ACS by recovering waste heat from engine exhaust and the hot water generated during engine cooling. However, these has mainly targeted each heat source separately without considering the possibility of combining both to supply the system at the same time. This manuscript explores the possibility of running ACS by capturing waste heat from vehicle engine exhaust and placing an extra heat exchanger between the generator and Primary heat exchanger. This additional heat exchanger extracts heat from the cooling water of the engine with an intent to lessen the thermal burden on the generator and, thereby, increase the efficacy of the system.

## 3.Absorption cooling system

### 3.1. System description

The lithium bromide-water ACS is a complex-high-efficient system with the intention to use the engine waste heat from the exhaust to provide eco-friendly cooling as illustrated in Fig 1. The ACS system has several major subcomponents through which the desired result of cooling is achieved in a very efficient manner. The generator inlet, state (4), is the central part of the system in which the $LiBr - H_2O$ is heated by the car exhaust waste heat. This heating process makes the water vaporized and separate from $LiBr - H_2O$. At the generator more heat is to transfer water vapor to state (8). At the condenser the heat is rejected and liquid water is formed, state (9). The liquid water leaves the expansion valve 2, state (9), then goes to the evaporator, state (10). At the evaporator, the heat is taken from the space or the environment to be cooled and that lead to convert the liquid to vapor. This absorbed heat cools the contents of the evaporator and

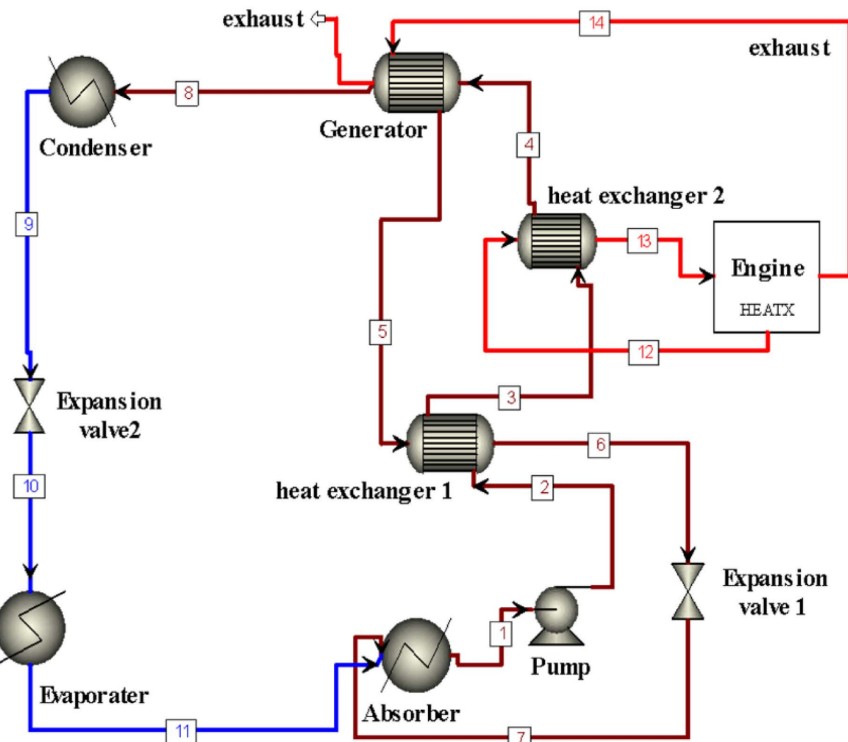

**Fig 1. A schematic view of ACS.**

gives the all-important cooling effect. After that, the generated water vapor is taken to the absorber, state (11), where it is re-absorbed in the concentrated LiBr solution. The absorption process releases heat with which the heat transfer in the cycle is rejected to the outer environment. The solution enters heat exchanger (1), which is situated between the absorber and generator in order to enhance the COP of the system. Heat exchanger (1) facilitates the transfer of heat from the hot solution departing the generator to the cooler solution entering it, thereby reducing the necessity for additional heat to be applied to its solution during reheating. Furthermore, the system incorporates an auxiliary heat exchanger state (3–4), which utilizes the heat from the cooling water of the car engine to increase the temperature of the solution by 10°C before enter the generator. This process further enhances the efficiency of the system by eliminating waste heat from being treated as wasted. Along with connecting pipes and supporting elements that help heat and fluid transfer within the system, a pump is added to move the fluid from the absorber to the generator. The purpose of this ACS is to maximize the utilization of energy resources and to regulate the environmental implications of emissions through conversion of wasted thermal energy into useful cooling.

### 3.2. Model setup

Previous studies have used various programs to simulate the operation of the ACS, including EES [47], Aspen [44], and FORTRAN software [34]. In this manuscript, MATLAB is used due to its clear technical language. MATLAB allows designing a mathematical model for ACS, particularly, in the thermal fluid field. The input parameters and the specified data which are used in the simulation are illustrated in Table 1. These parameters allow to calculate the enthalpy at all cycle states, the mass flow rate of the solution (in case of weak or strong) and the refrigerant ($\dot{m}_{ws}$, $\dot{m}_{ss}$, and $\dot{m}$), theconcentrations for the strong solution ($X_{ss}$) and the weak solution ($X_{ws}$), the pressure at all system states, and finally, the COP of the system.

### 3.3. Theoretical analysis

Energy balance analysis for the ACS has been suggested to be done by thermodynamic models. The model has however been developed from MATLAP software. Fig1 illustrates components/ state points of single-effect LiBr–water ACS, and waste heat of engine coupled with the generator of the system. Table 1 Summarizes the fixed limitation data utilized in the simulation.

   The following assumptions are made to derive the governing equations and develop the thermodynamic model for the present study. Similar studies use some of these assumptions [35–37].

1. The ACS is in a steady state.

2. Amplitude of the temperature is not significant in the main component of ACS.

3. The pressure is maintained same for Generator and condenser, as well as to the evaporator and absorber.

**Table 1. Fixed simulation data.**

| | |
|---|---|
| Generator temperature ($T_g$)(°C) | 90-120 |
| Absorber temperature ($T_a$) (°C) | 30-40 |
| Condenser temperature ($T_c$) (°C) | 30-40 |
| Evaporator temperature ($T_e$) (°C) | 10-20 |
| The evaporator cooling capacity (kw) | 5 |
| Solution heat exchanger (SHE.1) effectiveness | 0.5-0.95 |
| Heat exchanger (SHE.2) inlet temperature (°C) | 70 |
| SHE.2 effectiveness | 0.6 |

4. Refrigerant and solution expansion valves work in adiabatic process.

5. With reference to heat transfer and pressure drop to the environment, they are insignificant in all the components and the connecting tubes.

6. The fluid leaves the condenser as saturated liquid, state (9), and leaves the evaporator as a saturated vapour, state (11).

7. The cold medium of the server's refrigeration circuit moves heat to the evaporator.

### 3.3.1. Mass and energy at every element.

Mass and energy balance principles will be implemented on each component and subsequently on the entire system. As follows are the general forms of these equations:

$$\frac{dm}{dt} = \sum_{in} \dot{m} - \sum_{out} \dot{m} \tag{1}$$

$$\frac{dE}{dt} = \dot{Q} - \dot{W} + \sum_{in} \dot{m}h - \sum_{out} \dot{m}h \tag{2}$$

If the temperature at the condenser as well as that at the evaporator is known, the pressures corresponding to these temperatures can be determined by using the following empirical formula, which is derived by [38].

$$p = \exp\left(9.48654 + \frac{3892.7}{42.6776 - T}\right) \tag{3}$$

where, $T$ (K) and $p$ (MPa)

The provided equations below determine the concentration of LiBr in the strong and weak solutions, as the temperatures of the absorber, evaporator, generator, and condenser are known, Kaita [39].

$$X_1 = X_{ws} = \frac{49.04 + 1.125 T_a - T_e}{134.65 + 0.47 T_a} \tag{4}$$

$$X_5 = X_{ss} = \frac{49.04 + 1.125 T_g - T_c}{134.65 + 0.47 T_g} \tag{5}$$

where

$T$ (°C) is the temperature, and $X$ denotes absorbent concentration.

The energy balance of evaporator ($Q_e$) is given by:

$$Q_e = \dot{m}(h_{11} - h_{10}) \tag{6}$$

where,

$Q_e$ is an input value.

($h_{10}, h_{11}$) are enthalpies of saturated liquid and saturated vapour of the refrigerant in (kJ/kg), which can be can be calculated by the following equations:

$$h_{11} = 2501 + (1.88 * T_e) \tag{7}$$

$$h_{10} = 4.19 * T_c \tag{8}$$

The mass flow rate of the refrigerant is calculated using equation (6).

The energy balance of condenser ($Q_c$) is calculated from the following equations:

$$Q_c = \dot{m}(h_8 - h_9) \tag{9}$$

$$h_8 = 2501 + (1.88 * T_c) \tag{10}$$

$$h_9 = 4.19 * T_c \tag{11}$$

The energy balance of SHE.1

$$Q_{SHE.1} = \dot{m}_{ws}(h_3 - h_2) = \dot{m}_{ss}(h_6 - h_5) \tag{12}$$

$$\varepsilon_1 = \frac{(T_g - T_6)}{(T_g - T_a)} \tag{13}$$

where, $(\varepsilon_1)$ is the effectiveness of heat exchanger (input data).
Equation (13) is used to calculate the outlet temperature of SHE.1 $(T_6)$.
The specific heat of the weak solution can be calculated by:

$$C_{ws} = 4.184 * \left(1.01 - 1.23\, X_{ws} + 0.48\, X_{ws}^2\right) \tag{14}$$

The specific heat of the strong solution can be calculate using followings relation:

$$C_{ss} = 4.184 * \left(1.01 - 1.23\, X_{ss} + 0.48\, X_{ss}^2\right) \tag{15}$$

$$T_3 = T_a + \left[\varepsilon_1 * \left(\frac{X_{ws}}{X_{ss}}\right) * \left(\frac{C_{ss}}{C_{ws}}\right) * (T_g - T_a)\right] \tag{16}$$

The energy balance of SHE.2

$$Q_{HX.2} = \dot{m}_{ws}(h_4 - h_3) = \dot{m}_{wa} * C_{pw}(T_{12} - T_{13}) \tag{17}$$

where,
$(h_3, h_4)$ are the enthalpies of LiBr-water, $(T_{12}, T_{13})$ are the temperatures in (°C) of inlet and outlet hot water which comes from engine coolant water $(\dot{m}_{wa})$ is the mass flow rate of engine cooling water in (kg/s), and $C_{pw}$, is the specific heat of water in $(kJ/kg.K)$

$$\varepsilon_2 = \frac{Actual\ heat\ transfer\ rate\ (Q)}{maximum\ possible\ heat\ teansfer\ rate\ (Q_{max})} \tag{18}$$

$$Q_{max} = \dot{m}_{wa} * Cp_{wa} * (T_{12} - T_3) \tag{19}$$

where $(\varepsilon_2)$ the heat exchanger effectiveness
Absorber is given by the following equations:

$$Q_a = \dot{m}_{11}h_{11} + \dot{m}_7 h_7 - \dot{m}_1 h_1 \tag{20}$$

$$h_{11} = 2501 + (1.88 * T_e) \tag{}$$

Circulation ratio $(\lambda)$ can be calculate from following equation [40]:

$$\lambda = \frac{\dot{m}_{ss}}{\dot{m}} \tag{21}$$

$$\lambda = \frac{X_{ws}}{X_{ss} - X_{ws}} \tag{22}$$

where,

$$\dot{m}_{ws} = \dot{m}_{ss} + \dot{m} \tag{23}$$

The strong and weak mass flow rate solutions can be determined, with knowing refrigerant mass flow rate ($\dot{m}$), using equations 21–23.

The enthalpies of the solution at (state 1) and (state 5) are calculated using the standard formulations devised by the aforementioned individuals, based on the data obtained for the temperatures and concentrations, Kaita [39].

$$h(T, X) = (A_0 + A_1 X) \, T + 0.5 \, (B_0 + B_1 X) \, T^2 + \left(D_0 + D_1 X + D_2 X^2 + D_3 X^3\right) \tag{24}$$

where, (20 ≤ $T$ ≤ 210°C), and (40 ≤ X ≤ 65 wt.%),

| | | |
|---|---|---|
| $A_0 = 3.462023$ | $B_0 = 1.3499e^{-3}$ | $D_0 = 162.81$ |
| $A_1 = -2.679895e^{-2}$ | $B_1 = -6.55e^{-6}$ | $D_1 = -6.0418$ |
| | | $D_2 = 4.53488e^{-3}$ |
| | | $D_3 = 1.2053e^{-3}$ |

Now we can calculate the enthalpy of point (7, 1) by using above equation and substance in equation (20) to calculate ($Q_a$).

The value of heat addition rate in the generator ($Q_g$) is calculated as follows:

$$Q_g = \dot{m}_8 h_8 + \dot{m}_5 h_5 - \dot{m}_4 h_4 \tag{25}$$

$$h_7 = 2501 + (1.88 * T_e)$$

($h_5$, $h_4$) Can be calculated by using equation (24)

$$\dot{m}_4 = \dot{m}_8 + \dot{m}_5 \tag{26}$$

$$\dot{m}_4 = \dot{m}_{ws} \tag{27}$$

The expansion valve can be simplified, by taking into account assumptions (1), (3), and (4), as follows:

$$\dot{m}_9 = \dot{m}_{10} = \dot{m} \tag{28}$$

$$h_9 = h_{10} \tag{29}$$

$$h_{10} = 4.19 * T_c \tag{30}$$

The pressure of reducing valve is calculated as follows:

$$\dot{m}_6 = \dot{m}_7 = \dot{m}_{ss} \tag{31}$$

$$h_6 = h_7 \tag{32}$$

The enthalpy at point (6) is calculated using equation (24).

The density of strong solution at state (1) should be known in order to define the work done by the pump, and it can be estimated using the equation below [41]:

$$\rho = 1145.36 + 470.84 X_{ws} + 1374.79 X_{ws}^2 - (0.333393 + 0.571749 X_{ws}) \, T \tag{33}$$

The work done by the pump is [36]:

$$W_p = \frac{\dot{m}_{ws}(p_c - p_e)}{\rho_1}$$

(34)

where $(p_c - p_e)$ is the difference in pressure (Pa) between the evaporator and the condenser

**3.3.2. The performance of ACS.** The COP is typically employed to evaluate the performance of refrigeration systems. As the energy input of the system is divided by the useful effect produced, where this parameter is defined as:

$$COP = \frac{Q_e}{Q_g + Q_{HX.2} + W_p}$$

(35)

**3.3.3. Reduce in emission.** The reduction of emissions from the vehicle's exhaust is calculated based on the assumptions that the waste heat in the exhaust is 30 kW, the COP of the proposed ACS is 0.79, and the required cooling capacity is 5 kW. The emission reduction values are estimated using thermodynamic equations and the emission factors of each gas, as shown in Table 2 [42,43].

where, $E_f$ is the emission factor

$$Ein_{used} = \frac{Q_e}{COP}$$

(36)

$$E_{CO2} = Ein_{used} * Ef_{CO2}$$

(37)

$$E_{SOx} = Ein_{used} * Ef_{SOx}$$

(38)

$$E_{NOx} = Ein_{used} * Ef_{NOx}$$

(39)

$$CO2_{reduction} = (E_{in} * Ef_{CO2} - E_{CO2}) * time\_hours$$

(40)

$$SOx_{reduction} = (E_{in} * Ef_{SOx} - E_{SOx}) * time\_hours$$

(41)

$$NOx_{reduction} = (E_{in} * Ef_{NOx} - E_{NOx}) * time\_hours$$

(42)

## 4. Result and discussion

The results of the proposed model were compared to those of the Somers et al. [44] who employed (Aspen) software, as well as a mathematical model by Herold et al. [47] using (EES). Table 3 illustrates that the model results of this study and Aspen [44] which are in a good agreement. These discrepancies are likely attributed to differences in the calculation of refrigerant properties, solution specific heat, and density. Additionally, the experimental results of Florides et al. [45] and Patel et al. [46] were compared to the proposed model. The differences in COP values are in the range (4.9 to 8.23) %. For comparison, both, the study model results and the experimental results have been listed in Tables 4, and 5.

**Table 2. Emission factors.**

| $E_f$ | Value | Unit |
|---|---|---|
| $Ef_{CO2}$ | 0.25 | $\frac{Kg.\ CO_2}{KWh}$ |
| $Ef_{SOx}$ | 0.002 | $\frac{Kg.\ SOx}{KWh}$ |
| $Ef_{NOx}$ | 0.005 | $\frac{Kg.\ NOx}{KWh}$ |

**Table 3. Comparison between the proposed model and a theoretical result obtained from [47] and [44].**

| Parameter | Data type | Symbol unit | Herold et al. [47] | Somers et al. [44] | Present study | Dis-crep-ancy (%) |
|---|---|---|---|---|---|---|
| Generator temperature | Input | $T_g(°C)$ | 89.9 | 89.9 | 89.9 | -- |
| Evaporator temperature | Input | $T_e(°C)$ | 1.3 | 1.3 | 1.3 | -- |
| Condenser temperature | Input | $T_c(°C)$ | 40.2 | 40.2 | 40.2 | -- |
| Absorber temperature | input | $T_a(°C)$ | 32.7 | 32.7 | 32.7 | -- |
| SHE.1 effectiveness | input | $\varepsilon_1$ | 0.64 | 0.64 | 0.64 | -- |
| Evaporator cooling capacity | input | $Q_e(KW)$ | 10.772 | 10.772 | 10.772 | -- |
| Low pressure | Output | $P_e$ | 0.673 | 0.6715 | 0.6615 | 1.48 |
| High pressure | Output | $P_c$ | 7.445 | 7.4606 | 7.4381 | 0.3 |
| Concentration of weak solution | Output | $X_{ws}$ | 56.7 | 57.4 | 56.34 | 1.84 |
| Concentration of strong solution | Output | $X_{ss}$ | 62.5 | 62.57 | 62.16 | 0.65 |
| Generator load | Output | $Q_g(kw)$ | 14.952 | 14.952 | 14.588 | 2.43 |
| Condenser load | Output | $Q_c(kw)$ | 11.427 | 11.432 | 11.109 | 2.82 |
| Absorber load | Output | $Q_a(kw)$ | 14.297 | 13.923 | 14.246 | 2.32 |
| COP | Output | $cop$ | 0.720 | 0.738 | 0.7384 | 0.05 |

**Table 4. A comparison between the experimental results and the proposed model.**

Parameters: $T_g = 90 \,°C$, $T_e = 6 \,°C$, $T_c = 44.3 \,°C$, $T_a = 34.9 \,°C$, effectiveness of SHE.1=0.522

| Component | Florides et al. [45] Q (kW) | Present study | Discrepancy (%) |
|---|---|---|---|
| Generator | 14.2 | 11.970 | 15.7 |
| Condenser | 10.78 | 10.309 | 4.37 |
| Absorber | 13.42 | 13.687 | -1.92 |
| Evaporator | 10 | 10 | -- |
| COP | 0.704 | 0.7385 | 4.9 |

**Table 5. A comparison between the experimental results and the proposed model.**

Parameters: $T_g = 80 \,°C$, $T_e = 13 \,°C$, $T_a = T_c = 40 \,°C$, effectiveness of SHE.1=0.5

| Component | Patel et al. [46] Q (kW) | Present study | Discrepancy (%) |
|---|---|---|---|
| Generator | 200.8 | 182.388 | 9.17 |
| Condenser | 158.56 | 143.013 | 9.8 |
| Absorber | 195.84 | 182.388 | 6.68 |
| Evaporator | 140 | 140 | -- |
| COP | 0.692 | 0.749 | 8.23 |

**Table 6. Quantitative comparison of the proposed model with the Ammar and Seddiek [27].**

| Parameter | Data type | Symbol unit | Ammar and Seddiek [27] | Present study | Discrepancy (%) |
|---|---|---|---|---|---|
| Generator temperature | Input | $T_g(°C)$ | 90 | 90 | -- |
| Evaporator temperature | Input | $T_e(°C)$ | 6.7 | 6.7 | -- |
| Condenser temperature | Input | $T_c(°C)$ | 30 | 30 | -- |
| Absorber temperature | input | $T_a(°C)$ | 30 | 30 | -- |
| SHE.1 effectiveness | input | $\varepsilon_1$ | 0.7 | 0.7 | -- |
| Evaporator cooling capacity | input | $Q_e(KW)$ | 250 | 250 | -- |
| Concentration of weak solution | Output | $X_{ws}$ | 51 | 51.1 | 0.196 |
| Concentration of strong solution | Output | $X_{ss}$ | 68 | 67.89 | 0.161 |
| Generator load | Output | $Q_g(kw)$ | 312 | 295.17 | 5.4 |
| Condenser load | Output | $Q_c(kw)$ | 256 | 254.58 | 0.554 |
| Absorber load | Output | $Q_a(kw)$ | 298 | 297.19 | 0.278 |
| COP | Output | $COP$ | 0.827 | 0.829 | 0.241 |
| $CO_2$ reduction | Output | ton/year | 387 | 288.76 | 25.385 |
| SOx reduction | Output | ton/year | 2.46 | 2.310 | 6.097 |
| NOx reduction | Output | ton/year | 6.48 | 5.775 | 10.879 |

Table 6 shows a quantitative comparison of our work with Ammar and Seddiek [27]. Our investigation increased the COP by 0.241%, although the results were mostly similar to Ammar's. This is related to the second heat exchanger which is lowered the generator load by 5.4%. In addition, the difference in gas emission reduction between the two investigations is also affected by the engine type, where a gasoline engine is used in this study, while Ammar and Seddiek [27] were used diesel. The emissions of gasoline engine less than diesel engines, and that explains the disparity in emission reductions between the two studies.

## 4.1. Effect of generator temperature on ACS

The increase in Generator temperature makes it necessary to provide additional heat energy for the lithium bromide to produced water vapor in the generator. Although this enhances the heat transfer capability, it results in relative rise in temperatures beyond which the total thermal coefficient of performance begins to drop or remains constant at COP.

At higher absorber temperatures, there is a decrease in the rate of water vapor absorption by the lithium bromide solution as shown in Fig 2A. This is because the absorption process is endothermic, which leads to a decrease in the solution ability to absorb vapor at high temperatures. Therefore, the rate at which water vapor is produced by the generator will be reduced since the concentration of the solution is small. As a consequence, COP is reduced to 0.81 and 0.77 when the generator temperature was 90 °C and the absorber temperature were 30 °C and 40°C. respectively.

A rise in condenser temperature, as depicted in Fig 2B, leads to reduce the heat rejection from the condenser to the surroundings. The reduction in the temperature differential between the condenser and the environment means that the rate at which the water vapor is condensing will be affected. Consequently, the amount of water vapor which is taken to the evaporator to perform the cooling action is small, which in turn reduces the COP. From Fig 2B it can be seen that when condenser is at 30 °C and 40 °C, the COP obtained is 0.825 and 0.79, respectively, when the generator temperature is 90°C.

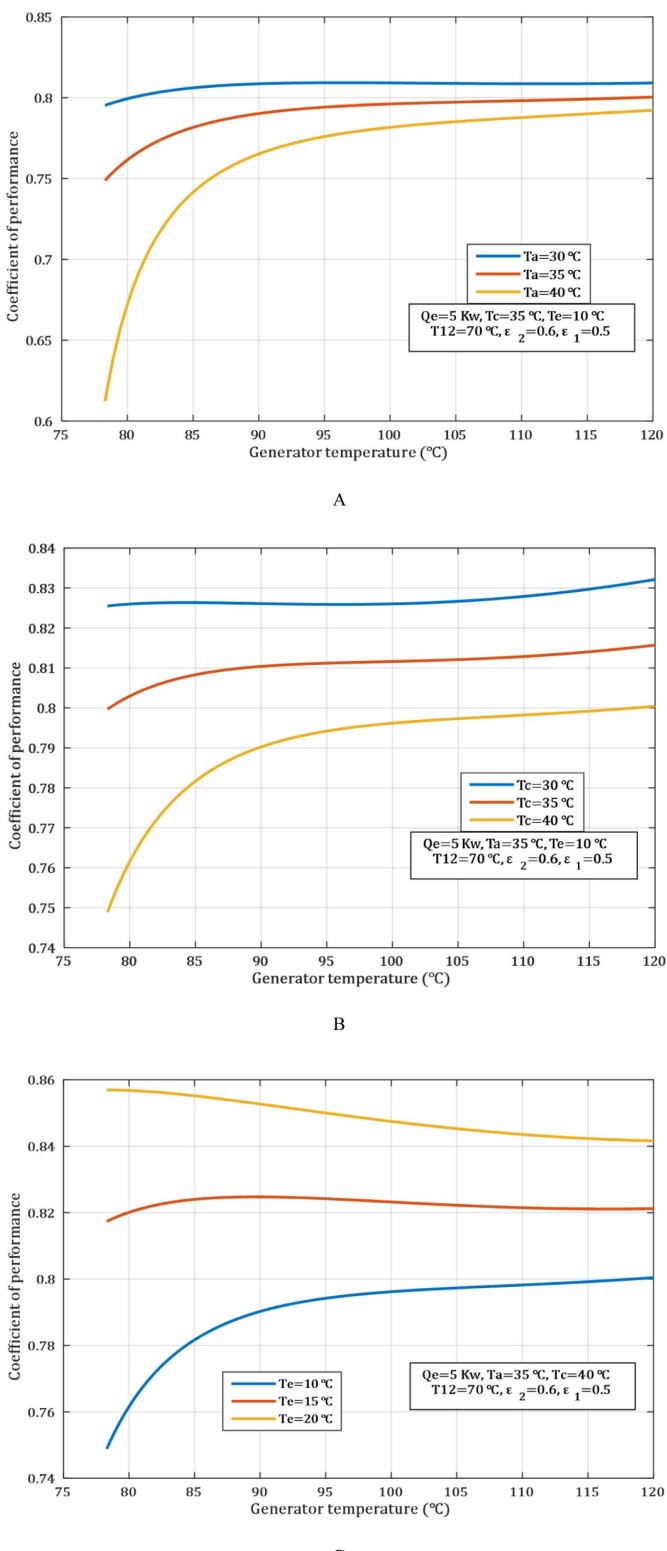

**Fig 2. The impact of generator temperature on COP.** (A) absorber temperature. (B) condenser temperature. (C) evaporator temperature.

Any increase in evaporator temperature is also lead to increase the differential between the evaporator and the media being cooled such as air or chilled water. This change in temperature reduces heat transfer time, thereby increasing the rate of cooling and the amount of heat to be transferred in the evaporator. As a result, the thermal load transfer through the system is increased, as well as the COP of the system. From the observations made, it can be seen that when the absorber temperature is 30°C and 40°C, the COP was 0.81 and 0.77 respectively, at 90°C of generator temperature, as shown in Fig 2C.

In an ACS using $LiBr-H_2O$, the circulation ratio ($\lambda$) is the mass of $LiBr$ solution circulated in the system relative to the mass of water vapor generated in the LiBr solution within the generator. The circulation ratio diminishes with the increase of the generator temperature and the decrease of the condenser and absorber temperatures. The efficiency of the process whereby water is separated from lithium bromide solution is enhanced by raising the temperature of the generator. This results in an increase in the concentration of the $LiBr$ solution in the absorber, and that is to a limited extent as discussed below. When the efficiency of the separation in the generator has been improved, the circulation of lithium bromide solution to generate the same quantity of water vapor decreases, and hence circulation ratio decreases too.

In Fig 3A, lowering the condenser temperature, also helps in enhancement of concentration of the water vapor removal. This reduction of pressure in the system reduces the intensity of the system to separate the water vapor from the generator. Thus, it becomes possible to achieve a lower flow rate that is necessary for circulating the solution. Fig 3B indicates that lowering absorber temperatures is improving the rate at which water vapour is absorbed into the lithium bromide solution. This means that to dissolve the amount of water vapor produced into the solution, less volumed is required and therefore the circulation ratio decreases. As the water concentration increases in the solution inside the absorber, the amount of solution to be circulated within the system reduces because water vapor is better absorbed efficiently.

Fig 3C shows the circulation ratio is determined by a quantity of solution required for heat transportation for cooling. The higher the evaporator temperature, leads the system to achieve higher COP with a lower concentration of the solution. This is yielding a lower circulation ratio. Similarly, a low evaporator temperature implies a high cooling effect, but a larger volume of solution is used; thus, the large circulation ratio.

Before entering the generator, the solution is pre-heated with engine coolant water. This reduces the thermal load on the generator, and enables utilization of waste heat from the exhaust of the engine. This reduces dependency on exhaust heat, which makes the efficiency of the generator and the sustainability of the system to be more efficient. Therefore, one can minimize a large thermal fluctuation in the generator, thereby increasing the efficiency and reliability of the generator when its thermal load is minimized. This can be observed in Fig 4A which shows that the generator load was 5.8 kW, while in Fig 4B it was 6.24 kW under the same condition, such as the absorber and generator temperatures, which are 30°C and 80°C, respectively.

## 4.2. Effect of heat exchangers effectiveness

The enhancement in the heat exchanger enhances the COP at low condenser temperatures due to low pressure and thermal load. When the SHE.1 effectiveness was 0.5 and 0.85, the COP values was 0.79 and 0.84, respectively, at a condenser temperature of 40°C, under the same conditions as shown in the Fig 5. Although the effect is favorable at high condenser temperatures, it is not as distinctive as at lower temperatures. This is because high system pressure conditions limit the available thermal expansion. Improving heat exchanger efficiency is another way of increasing system efficiency and thereby reducing energy consumptions especially when the condenser is low.

Fig 6 shows the influence of the heat exchanger on the generator thermal load, which is clearly observed. It is clear from the Fig 6A that the thermal load of the generator absorbs drops as the heat exchanger's efficiency rises. Thus, this behaviour is also due to the fact that instead of having an additional heat exchanger, the heat is only transferred from the engine coolant water. The Fig 6B demonstrates the same trend but with the addition of a SHE.2. The presence of

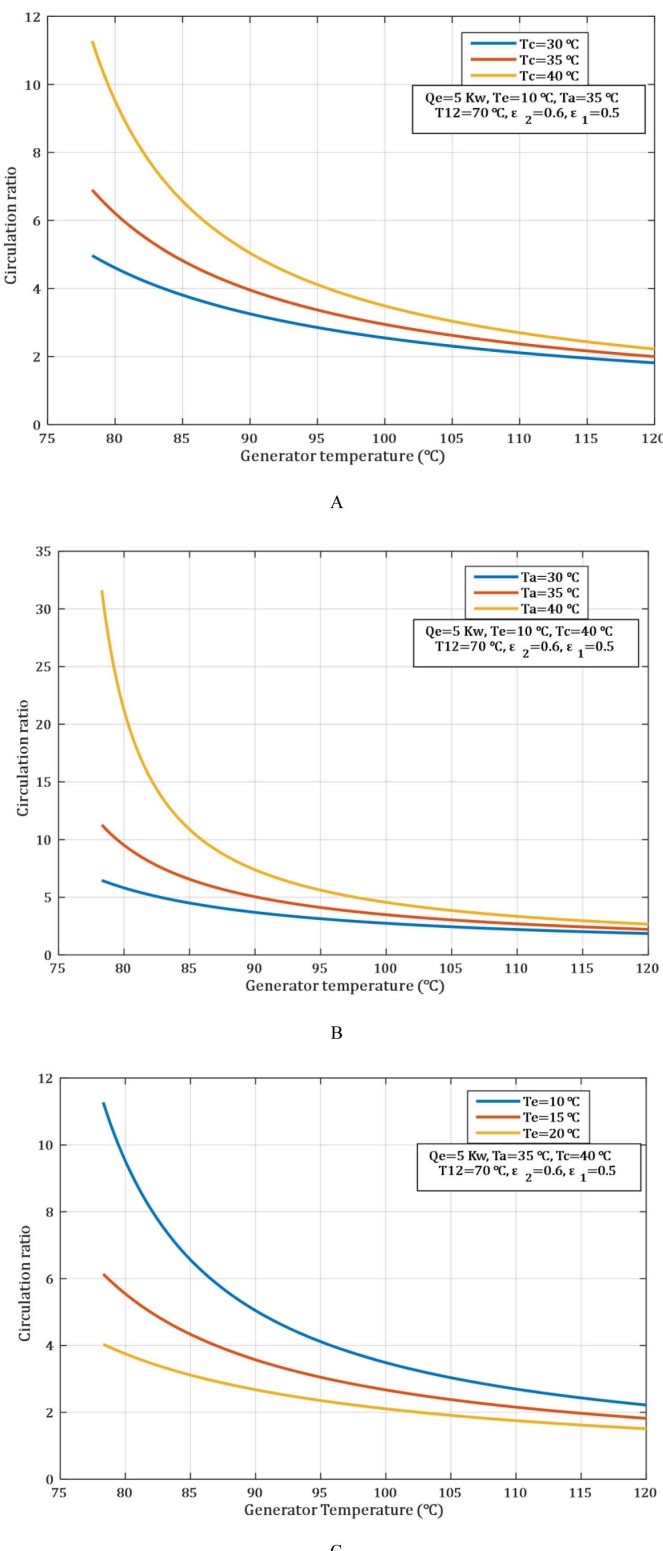

**Fig 3. Influence of generator temperature on circulation ratio.** (A) absorber temperature. (B) condenser temperature. (C) evaporator temperature.

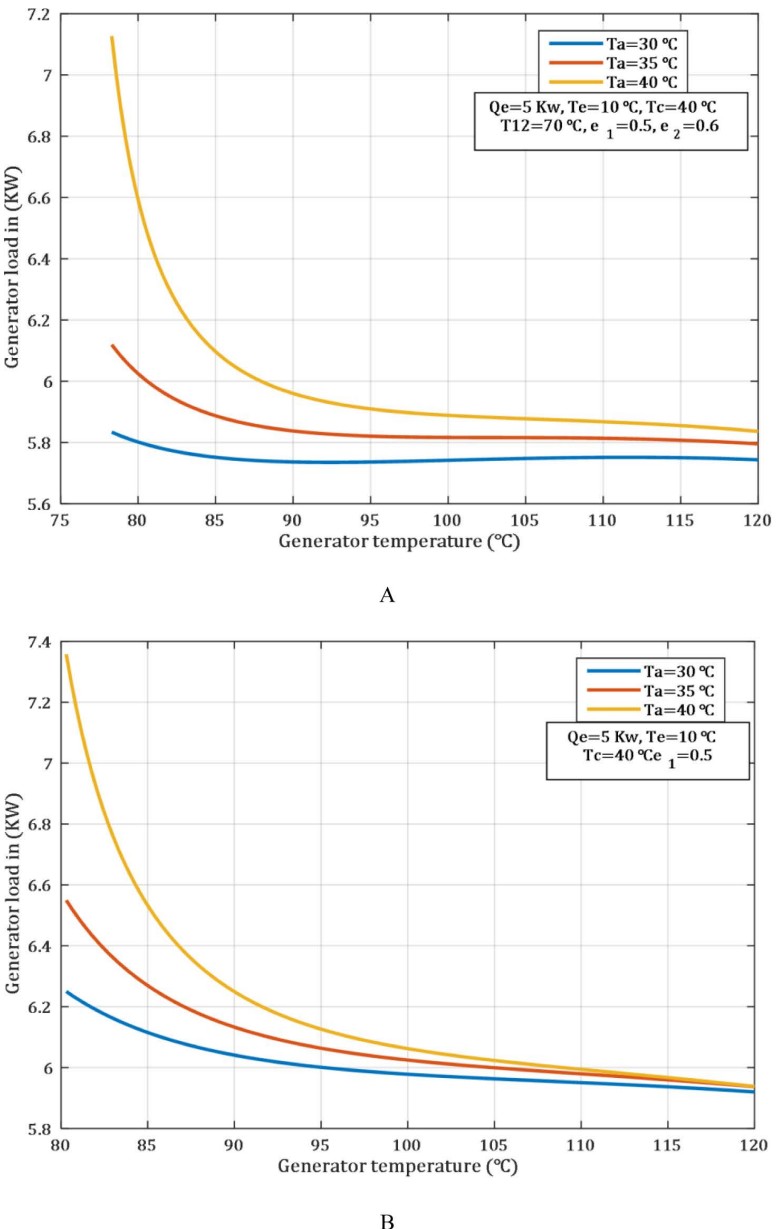

A

B

**Fig 4. Influence of generator temperature on generator load at various temperature of Ta.** (A) with SHE.2. (B) without SHE.2.

the SHE.2 leads to a further reduction in the thermal load consumed by the generator, resulting in a significant decrease compared to the first case at Fig 6A.

More heat transfer occurs between the two solutions when the heat exchanger between the strong solution, which returns to the generator, and the weak solution, going to the generator, becomes more efficient. This, in turn, reduces the demand of the absorber to deal with heat observed, as in Fig 7. In the weak solution, the strong solution becomes even more capable of accepting water vapour when the absorber temperature is lowered. Optimum absorption lowers the overall mass of the absorber and also minimizes heat produced during the absorption process.

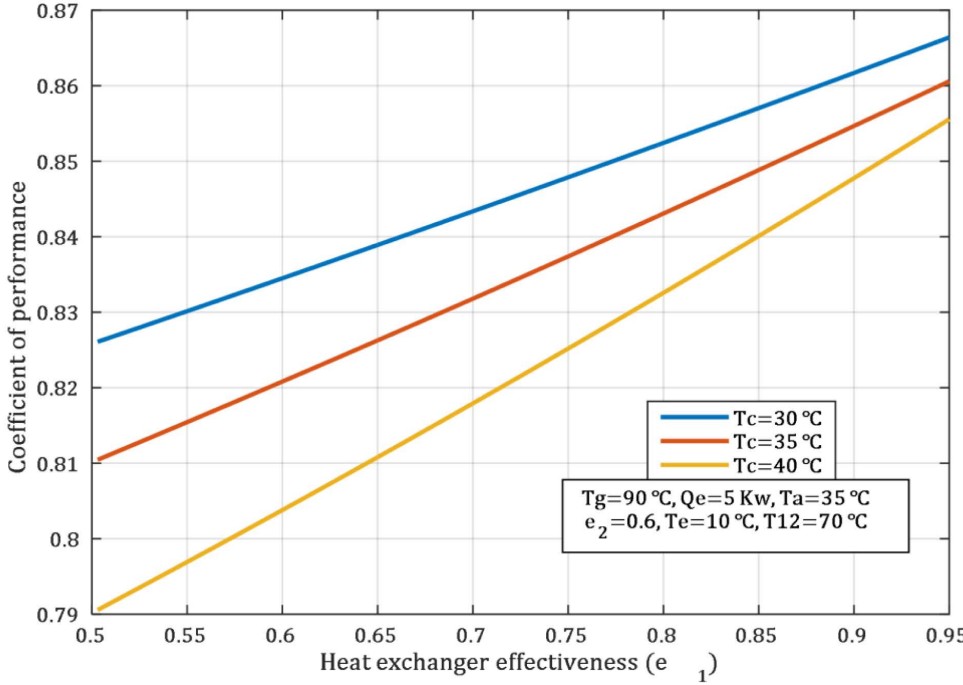

**Fig 5. Impact of heat exchanger effectiveness on COP.**

### 4.3. Effect of concentration solution on system

The impact of $X_{ws}$ and absorber temperature on the COP of an ACS is shown in the Fig 8. when concentration of weak solution increases the COP decrease, because of increased the thermal energy required in the generator to separate the concentrated solution. Finally, all these changes lead to reduce system efficiency and increase operational expenses.

When the concentration of the strong solution is high, the concentration difference between the strong solution and water vapor going into the absorber is large. This difference increases the possibility of the strong solution to pick up more water vapor in the absorber. At low absorber temperatures, the utilization of the strong solution is high since the absorption reaction of $LiBr - H_2O$ is more preferred at lower temperatures. When the concentration was 0.55 and 0.6, the COP was 0.75 and 0.8, respectively, at an absorber temperature of 30°C, as Fig 9. These values were obtained under the same conditions.

### 4.4. Emission reduction

However, as it is evident in the Fig 10, the exhaust heat from the car engine for the ACS has been showing progressive reductions in the emissions of $CO_2$, $SOx$, and $NOx$. The system reduces $CO_2$ emissions by approximately 1.58 kg per hour of operation, demonstrating its significant contribution to emissions reduction. The proposed system is essential in the reduction of $SOx$ and $NOx$ emissions, despite the fact that it is lower than the $CO_2$ level that was attained. Table 7 shows the amount of emission reductions for the three gases ($CO_2$, $SOx$, and $NOx$) per hour of operation. When the system is designed to run for longer durations, this leads to higher effectiveness in cutting emissions. Thus, it is raising the efficiency of the system. More improvements can be made to the system in order to lower emission levels further, thereby increasing the overall beneficial environmental effects.

Table 8 includes the list of thermodynamic properties and flow parameters of different state points in the thermodynamic system. Table 8 offers essential information such as temperature (°C) that refers to thermal status at a specific point;

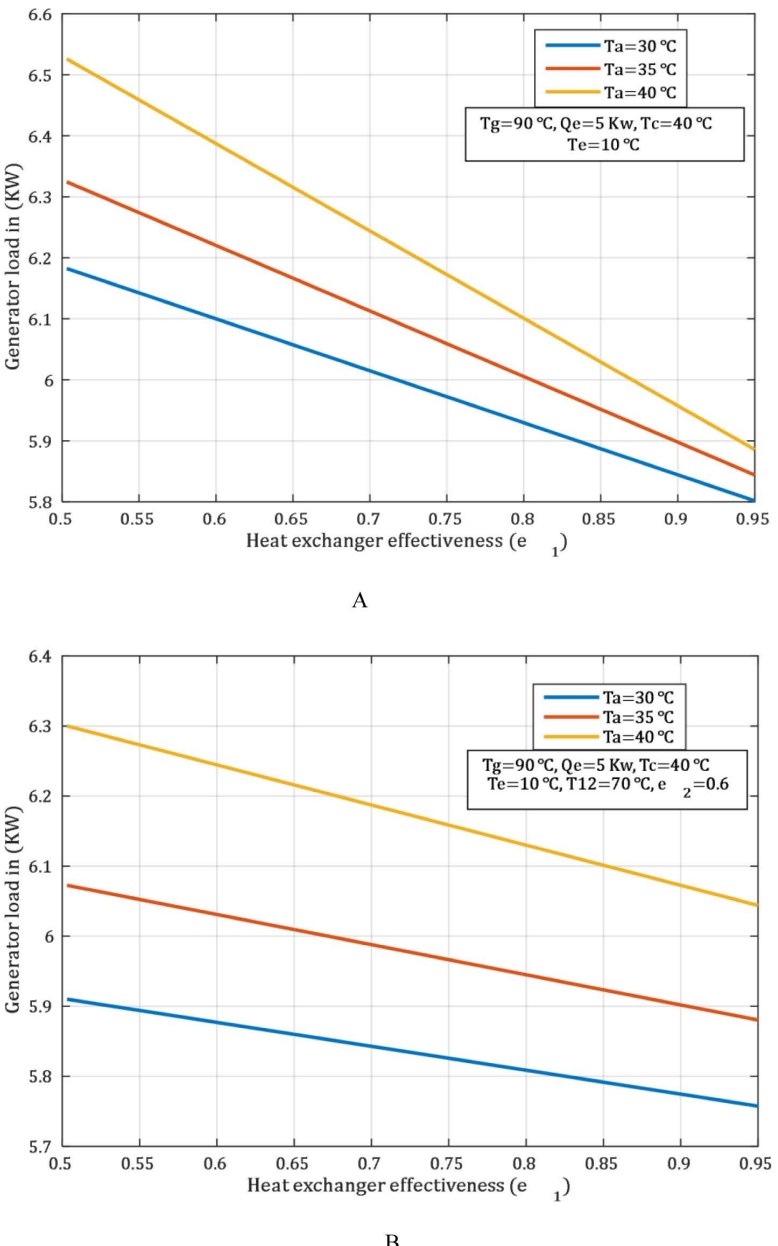

A

B

**Fig 6. Effect of effectiveness heat exchanger on generator load at various temperature of Ta.** (A) without SHE.2 and (B) with SHE.2.

enthalpy (kJ/kg) that shows the energy of working fluid. Pressure (kPa) means the fluctuations in the system pressure, whereas concentration (%) refers to the make-up of the working fluid, which is essential for absorption systems. Moreover, mass flow rate (kg/s) is used to describe the rate of the fluid substitute, and heat transfer rate (Q, kW) denotes energy interchange at various points. In Table 8, two values are given for the generator load. The first value at (6.027 kW) which shows the generator load without using the second heat exchanger while the second value at point (5.119 kW) which shows the generator load while using the second heat exchanger. Table 8 shows how the heat exchanger is effectively

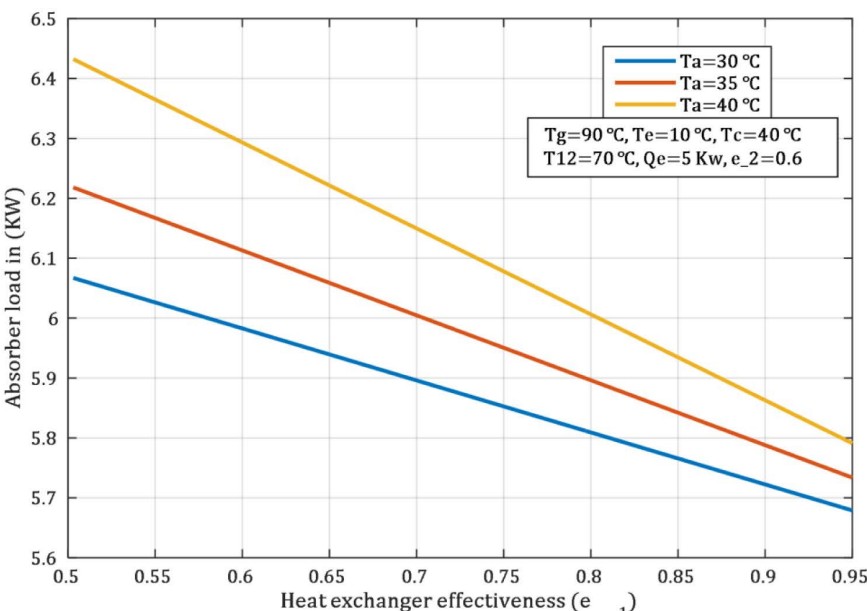

**Fig 7. Effect of effectiveness heat exchanger on Absorber load.**

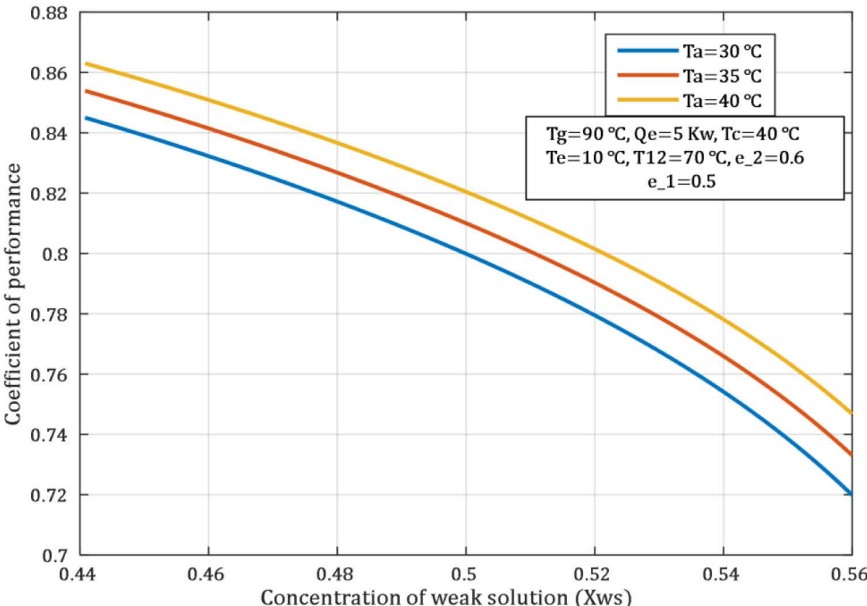

**Fig 8. Influence of concentration of weak solution on COP.**

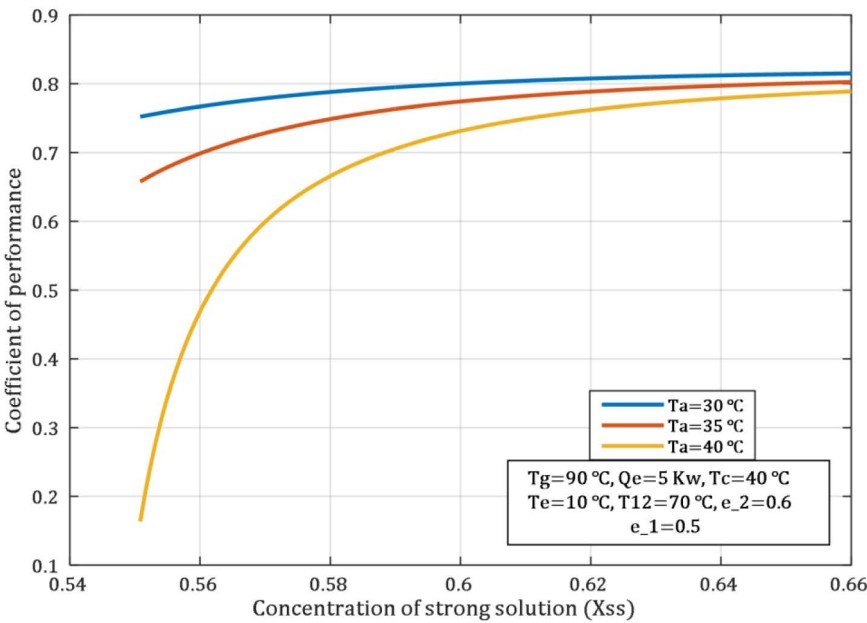

**Fig 9. Influence of concentration of strong solution on COP.**

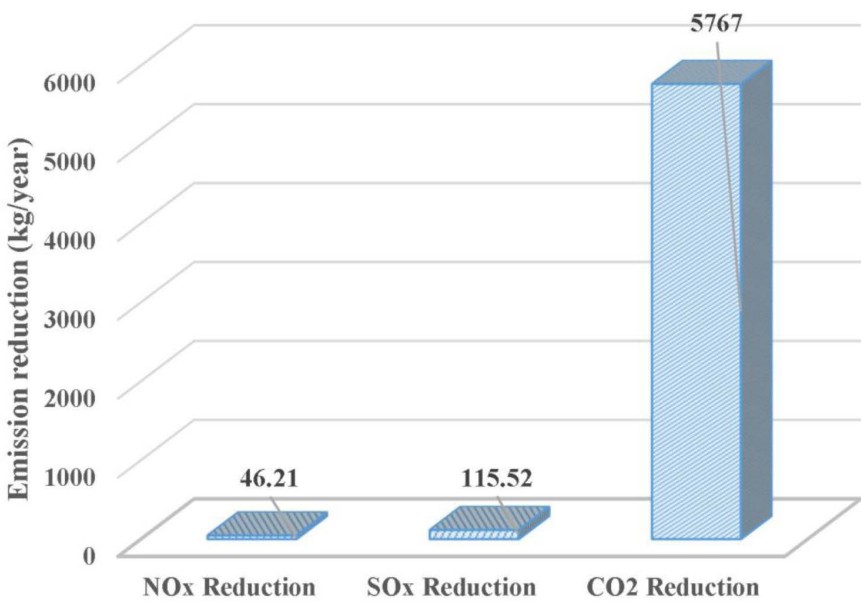

**Fig 10. Magnitude of emission reduction.**

**Table 7. Shows the amount of emission reductions.**

| Time (hours) | $CO_2$ Reduction (kg) | NOx Reduction (kg) | SOx Reduction (kg) |
|---|---|---|---|
| 0 | 0 | 0 | 0 |
| 1 | 1.58 | 0.03165 | 0.01266 |
| 2 | 3.16 | 0.06330 | 0.02532 |
| 3 | 4.74 | 0.09495 | 0.03798 |
| 4 | 6.32 | 0.12660 | 0.05064 |
| 5 | 7.90 | 0.15825 | 0.06330 |
| 6 | 9.48 | 0.18990 | 0.07596 |
| 7 | 11.06 | 0.22155 | 0.08862 |
| 8 | 12.64 | 0.25320 | 0.10128 |
| 9 | 14.22 | 0.28485 | 0.11394 |
| 10 | 15.80 | 0.31650 | 0.12660 |

**Table 8. Absorption cycle typical state point results from MATLAB models.**

| State point | From | To | Temperature °C | Enthalpy Kj/kg | Pressure Kpa | Concentration % | Mass flow rate Kg/s | Q (KW) |
|---|---|---|---|---|---|---|---|---|
| 1 | absorber | pump | 35 | 74.487 | 1.216 | 51.896 | 0.01269 | 6.218 |
| 2 | pump | SHE.1 | 35 | 74.487 | 7.359 | 51.896 | 0.01269 | – |
| 3 | SHE.1 | SHE.2 | 54.64 | 116.690 | 7.359 | 51.896 | 0.01269 | 0.535 |
| 4 | SHE.2 | Generator | 63.85 | 136.521 | 7.359 | 51.896 | 0.01269 | 0.251 |
| 5 | Generator | SHE.1 | 90 | 220.378 | 7.359 | 62.328 | 0.01057 | 6.072 5.119 |
| 6 | SHE.1 | Valve.1 | 62.5 | 170.970 | 7.359 | 62.328 | 0.01057 | – |
| 7 | Valve.1 | absorber | 62.5 | 170.970 | 1.216 | 62.328 | 0.01057 | – |
| 8 | Generator | condenser | 90 | 2576.2 | 7.359 | 0 | 0.00212 | – |
| 9 | condenser | Valve.2 | 40 | 167.600 | 7.359 | 0 | 0.00212 | 5.119 |
| 10 | Valve.2 | evaporator | 40 | 167.600 | 1.216 | 0 | 0.00212 | – |
| 11 | evaporator | absorber | 10 | 167.600 | 1.216 | 0 | 0.00212 | 5.000 |
| 12 | engine | SHE.2 | 70 | 293 | 101.32 | – | 0.2 | – |
| 13 | SHE.2 | Radiator | 65 | 270 | 101.32 | – | 0.2 | – |

used to enhance the COP of the system when the necessary thermal load for the generator is lowered. It offers a holistic insight into the performance, productivity, and thermal characterization of the system at various sections.

### 4.5. Sensitivity analysis

**4.5.1. Effect of generator temperature.** Table 9 assumed the following conditions to evaluate this effect:

The sensitivity and error analysis of the mathematical model are illustrated in Fig 11, which illustrates the impact of generator temperature on the COP. The elevated pressure reduces efficiency, and that causing the COP to decrease as temperature increase. It shows slight change in the coefficient, implying that the sensitivity analysis indicates higher stability at high temperatures. An error rate of less than 2.17% demonstrates the accuracy of the model. In general, the model functions satisfactorily and exhibits moderate sensitivity and dependable outcomes.

Fig 12 shows the sensitivity analysis of generator temperature in ACS directly compared with Florides et al. in terms of the COP under the same operating conditions as shown in the Table 9. Both studies result in a gradual reduction in

**Table 9. Assumed conditions.**

| | |
|---|---|
| Evaporator temperature ($T_e$) | 10°C |
| Absorber temperature ($T_a$) | 35°C |
| SHE.1 exit temperature, $T_3$ | 55°C |
| Evaporator capacity ($Q_e$) | 5 *KW* |
| Weak solution concentration ($X_{ws}$) | 52% |
| Strong solution concentration ($X_{ss}$) | 60% |
| Pressure in absorber and evaporator ($P_e$) | 1.216 *kpa* |
| SHE.2 effectiveness | 0.6 |

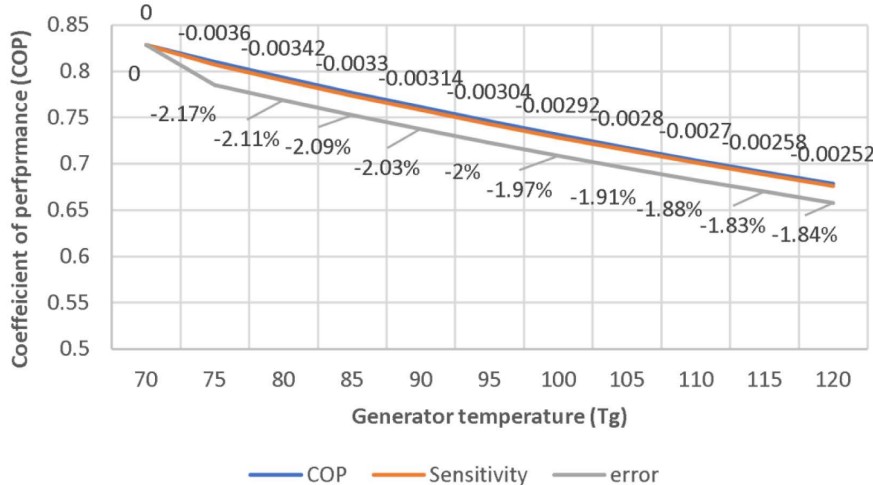

**Fig 11. Effect of generator temperature on COP.**

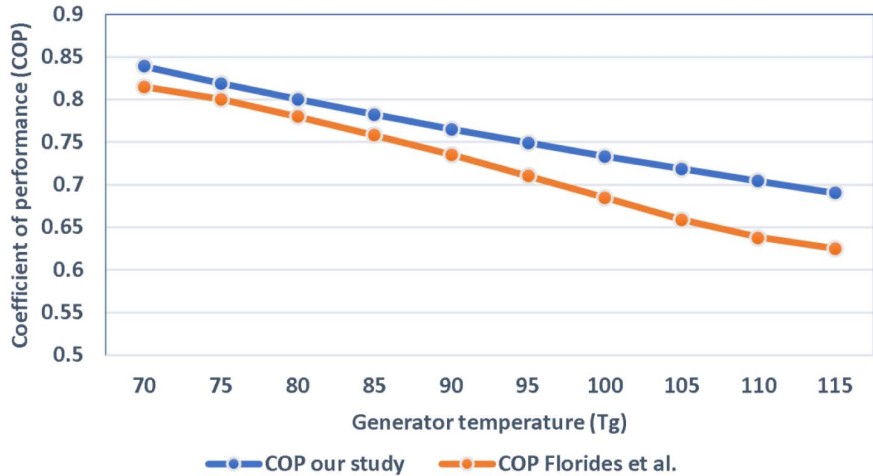

**Fig 12. Comparison COP of our study with Florides et al.**

COP with an increase in the generator temperature, and that indicates the system's sensitivity to changes in generator temperature. Findings of the current study are a 5–8% improvement in COP compared with the result of Florides et al., at generator temperatures between 70 and 115 °C. The slight difference reported came from the precision of the mathematical model and is, perhaps, from the mere slight variation of the calculation assumptions and the secondary thermal parameters.

**4.5.2. Effect of SHE.1 effectiveness.** Table 10 assumed the following conditions to evaluate this effect:

Fig 13 illustrates the degree at which the COP of the system is influenced by the effectiveness of the heat exchanger under the fixed conditions outlined in Table 10. The Fig 13 indicates that an increase in the effectiveness of the heat exchanger results in an increase in the COP. The COP was 0.765 at an effectiveness of 0.5 and 0.838 at 0.9. The Fig 13 also illustrates the degree to which the heat exchanger activities influence the temperatures entering and exiting the generator, as well as their influence on the generator load.

**4.5.3. Effect of weak solution concentration.** Table 11 assumed the following conditions to evaluate this effect:

Fig 14 shows the sensitivity analysis of the effect of changing the concentration of weak solution ($X_{ws}$) on the COP of the ACS, as the results show that with increasing concentration from 0.45 to 0.55, the COP decreases from 0.87 to 0.71

**Table 10. Assumed conditions.**

| | |
|---|---|
| Evaporator temperature ($T_e$) | 10°C |
| Absorber temperature ($T_a$) | 35°C |
| Condenser temperature ($T_c$) | 55°C |
| Evaporator capacity ($Q_e$) | 5 *KW* |
| Weak solution concentration ($X_{ws}$) | 52% |
| Strong solution concentration ($X_{ss}$) | 60% |
| Pressure in absorber and evaporator ($P_e$) | 1.216 *kpa* |
| Pressure in generator and condenser ($P_c$) | 7.359 *kpa* |
| SHE.2 effectiveness | 0.6 |

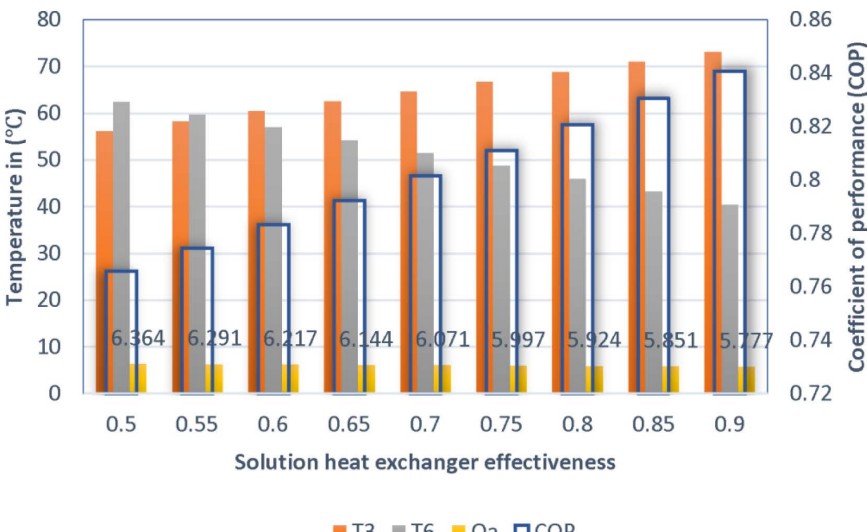

**Fig 13. Effect of SHE.1 effectiveness on COP at different temperature.**

**Table 11. Assumed conditions.**

| | |
|---|---|
| Evaporator temperature ($T_e$) | 10°C |
| Absorber temperature ($T_a$) | 35°C |
| Condenser temperature ($T_c$) | 55°C |
| Generator temperature ($T_g$) | 90 |
| Evaporator capacity ($Q_e$) | 5 *KW* |
| Strong solution concentration ($X_{ss}$) | 60% |
| Pressure in absorber and evaporator ($P_e$) | 1.216 *kpa* |
| Pressure in generator and condenser ($P_c$) | 7.359 *kpa* |
| SHE.1 effectiveness | 0.5 |
| SHE.2 effectiveness | 0.6 |

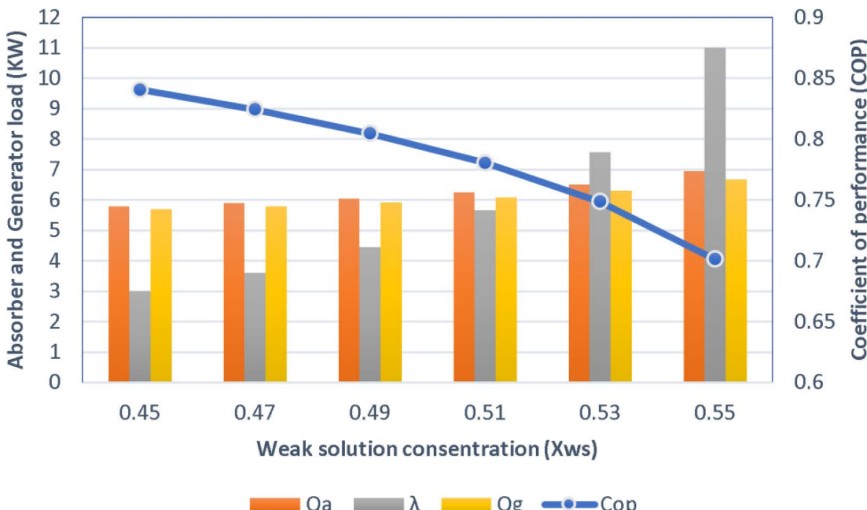

**Fig 14. Effect of weak solution concentration on COP, $Q_a$, $Q_g$, and $\lambda$.**

approximately, while the circulation ratio (λ) increases significantly. In contrast, thermal loads on the absorbent ($Q_a$) and generator ($Q_g$) show only a slight change, reflecting a low sensitivity of these loads to concentration changes compared to performance and flow rate. These results confirm the need to choose the optimal concentration of the solution to achieve high performance with minimal energy consumption.

## 5. Conclusion

The ACS was analyzed with respect to the contribution of operational parameters and the addition of heat exchangers to increase its efficiency.

- The results show that any changes in the temperatures of the generator, condenser and absorber will disturb the thermal balance and that leads to lower COP, while increasing the aperture of the generator and evaporator leads to improve energy transfer efficiency and increase the COP.

- The COP values were 0.77, 0.79, and 0.81 at absorber temperatures of 30°C, 35°C, and 40°C, respectively, under the same operating conditions. Similarly, the COP values were 0.79, 0.81, and 0.826 at condenser temperatures of 40°C,

35°C, and 30°C, respectively. Furthermore, the COP values increased to 0.79, 0.825, and 0.854 at evaporator temperatures of 10°C, 15°C, and 20°C, respectively, under the same operating conditions.

• Lowering the absorber and condenser temperatures together with raising the generator temperature improves the separation, condensation and absorption processes. Thus, the circulation ratio and the operation efficiency are improved.

• The results also indicated that increasing the $X_{ws}$ negatively affects the COP by reducing the absorber heat absorption capacity, increasing energy consumption, and raising pressure, while a higher $X_{ss}$ from the generator improves thermal efficiency and system stability at lower absorber temperatures.

• Improving the effectiveness of heat exchangers and lowering absorber temperatures reduces thermal load, increases the absorption rate, and boosts system efficiency.

• The addition of a secondary heat exchanger further reduced the thermal load on the generator by 4% to 7%, depending on the operating conditions. Specifically, with the secondary heat exchanger, the generator load was 5.84 kW, compared to 6.15 kW without the secondary heat exchanger under the same operating conditions. Thus, the importance of improving system performance and efficiency through integration of a secondary heat exchanger is demonstrated.

• The proposed system showed environmental benefits of 1.58 kg of $CO_2$ per hour, and of decreasing $SOx$ and $NOx$ emissions to levels below those required by the applicable regulations.

• Extended operation further increases these environmental benefits, and further optimization is capable of increasing emission reductions and efficiency nearly to the extent that the system has truly become a sustainable solution of utilizing waste heat. It helps in creating a clean environment, reducing fuel consumption and being energy efficient, which in turn serves our society by helping make it healthier as a result of lower chances for environmental damage.

## Nomenclature

| Name | Description | Name | Description |
|---|---|---|---|
| $h$ | Enthalpy (kJ/kg) | ACS | Absorption cooling system |
| $\dot{Q}$ | Heat quantity (kW) | DAR | Diffusion absorption system |
| $T$ | Temperature (K) | HCFCs | Hydrochlorofluorocarbons |
| $X$ | Concentration of LiBr | HCs | Hydrocarbon |
| $\dot{m}$ | Mass flow rate, (kg/s) | $CO_2$ | Carbon dioxide |
| $p$ | Pressure (kpa) | VARS | Vapour absorption refrigeration system |
| $\varepsilon$ | Effectiveness | NOx | Oxides of nitrogen |
| $C$ | Specific heat (kJ/kg) | SOx | Oxides of sulfur |
| $W_p$ | Power (kW) | **Subscripts** | |
| $\rho$ | Solution density | $a$ | absorber |
| $\lambda$ | Circulation ratio | $c$ | Condenser |
| **COP** | Coefficient of performance | $e$ | Evaporator |
| **Ef$_{CO2}$** | emission factor of $CO_2$ | $g$ | Generator |
| **Ef$_{NOx}$** | emission factor of $NO_x$ | $ss$ | Strong solution |
| **Ef$_{SOx}$** | emission factor of $SO_x$ | $ws$ | Weak solution |

## Author contributions

**Conceptualization:** Mohammed Qasim Shaheen, Salman Hashim Hammadi.

**Data curation:** Mohammed Qasim Shaheen.

**Formal analysis:** Mohammed Qasim Shaheen, Salman Hashim Hammadi.

**Investigation:** Salman Hashim Hammadi.

**Supervision:** Salman Hashim Hammadi.

**Writing – original draft:** Mohammed Qasim Shaheen.

**Writing – review & editing:** Mohammed Qasim Shaheen.

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
