## [Decision Letter · Decision Letter 0]

28 Feb 2025

PONE-D-25-03688A Sustainable Lithium Bromide-Water Absorption Cooling System Using Automobile Engine Waste Heat: Theoretical StudyPLOS ONE

Dear Dr. Shaheen,

Thank you for submitting your manuscript to PLOS ONE. After careful consideration, we feel that it has merit but does not fully meet PLOS ONE’s publication criteria as it currently stands. Therefore, we invite you to submit a revised version of the manuscript that addresses the points raised during the review process.

We look forward to receiving your revised manuscript.

Kind regards,

Joy Nondy, Ph. D.

Academic Editor

PLOS ONE

Reviewers' comments:

Reviewer's Responses to Questions

**Comments to the Author**

1. Is the manuscript technically sound, and do the data support the conclusions?

Reviewer #1: Partly

Reviewer #2: Yes

2. Has the statistical analysis been performed appropriately and rigorously? 

Reviewer #1: No

Reviewer #2: Yes

3. Have the authors made all data underlying the findings in their manuscript fully available?

Reviewer #1: Yes

Reviewer #2: Yes

4. Is the manuscript presented in an intelligible fashion and written in standard English?

Reviewer #1: No

Reviewer #2: No

5. Review Comments to the Author

Reviewer #1: This manuscript investigates the feasibility of utilizing a lithium bromide-water absorption cooling system (ACS) powered by waste heat from automobile engines. The study employs thermodynamic modeling and MATLAB simulations to evaluate system performance, particularly in terms of coefficient of performance (COP) and emission reductions. While the study is well-structured and methodologically sound, its not ready for publication in present form. Here is the list of doubts and questions:

1. The study is entirely theoretical. The authors should either provide experimental data to validate the model, or compare their results with published experimental studies to demonstrate the models reliability.

2 The manuscript references multiple previous experimental studies, however, it does not quantitatively compare its results with real-world data (e.g., COP values, temperature ranges, emission reductions). Given the large body of research on LiBr-H₂O systems, this omission weakens the credibility of the findings.

3. The novelty of the paper is not clear. The equations and modeling approach follow conventional thermodynamic principles without introducing significant new computational methodologies. The analysed temperature range is typical for an absorption system, and the system itself also looks conventional. Therefore, the authors are asked to enhance the explanation of why this paper should be published?

4. The manuscript does not include an error analysis. The authors should assess how sensitive the results are to input parameters, estimate potential uncertainties, and compare their findings with experimental data to ensure their accuracy

Reviewer #2: 1. The temperature should be clearly mentioned in the abstract and manuscript. Because only single temperature has been mentioned for generator, absorber, condenser and evaporator. However, the variation of temperatures is shown in result.

2. Define the objective of the study clearly. Rewrite your abstract.

3. Mention the boundary condition and limitation of the study in methodology.

4. Modify the figure 2. Axis titles are missing. Also cite the manuscript from which this study was validated. It is not an ethical way to write “another research”.

5. Show the standard deviation for bar chat graphs.

6. From figure 5-11, there are several sub-figures available. But, these figures are not mentioned in captions. So mention all the figures in figure caption.

7. Check the heading numbers of the manuscript.

8. In figure 11, SOx and NOx reduction is not clearly visible. So modify the figure11 in such a way that all data should clearly visible in manuscript.

9. Highlight the major findings of the study in the bullet point in conclusion section and also mention that how this study will help the society.

10. Check the grammatical errors in the manuscript.

6. PLOS authors have the option to publish the peer review history of their article (what does this mean? ). If published, this will include your full peer review and any attached files.

**Do you want your identity to be public for this peer review?** For information about this choice, including consent withdrawal, please see our Privacy Policy .

Reviewer #1: No

Reviewer #2: No

---

## [Author Response · Author response to Decision Letter 1]

16 Apr 2025

We thank the reviewers and editor for their constructive feedback. All comments have been addressed in the revised manuscript. A detailed point-by-point response has been provided in the “Response to Reviewers” document. Revised and clean copies of the manuscript have been uploaded accordingly.

---

## [Decision Letter · Decision Letter 1]

2 May 2025

A sustainable lithium bromide–water absorption cooling system using automobile engine waste heat: Theoretical study

PONE-D-25-03688R1

Dear Dr. Shaheen,

We’re pleased to inform you that your manuscript has been judged scientifically suitable for publication and will be formally accepted for publication once it meets all outstanding technical requirements.

Kind regards,

Joy Nondy, Ph. D.

Academic Editor

PLOS ONE

Additional Editor Comments (optional):

Reviewers' comments:

Reviewer's Responses to Questions

**Comments to the Author**

1. If the authors have adequately addressed your comments raised in a previous round of review and you feel that this manuscript is now acceptable for publication, you may indicate that here to bypass the “Comments to the Author” section, enter your conflict of interest statement in the “Confidential to Editor” section, and submit your "Accept" recommendation.

Reviewer #1: All comments have been addressed

2. Is the manuscript technically sound, and do the data support the conclusions?

Reviewer #1: Yes

3. Has the statistical analysis been performed appropriately and rigorously? 

Reviewer #1: Yes

4. Have the authors made all data underlying the findings in their manuscript fully available?

Reviewer #1: Yes

5. Is the manuscript presented in an intelligible fashion and written in standard English?

Reviewer #1: Yes

6. Review Comments to the Author

Reviewer #1: I recommend the article for publication in its current form, as the authors have successfully addressed all comments and provided the necessary information to make the work complete.

7. PLOS authors have the option to publish the peer review history of their article (what does this mean? ). If published, this will include your full peer review and any attached files.

**Do you want your identity to be public for this peer review?** For information about this choice, including consent withdrawal, please see our Privacy Policy .

Reviewer #1: No

---

## [Editor Report · Acceptance letter]

PONE-D-25-03688R1

PLOS ONE

Dear Dr. Shaheen,

I'm pleased to inform you that your manuscript has been deemed suitable for publication in PLOS ONE. Congratulations! Your manuscript is now being handed over to our production team.

Kind regards,

on behalf of

Dr. Joy Nondy

Academic Editor

PLOS ONE